# WoW: Scaling Embodied Omni-World Model For Generalizable Manipulation Simulation

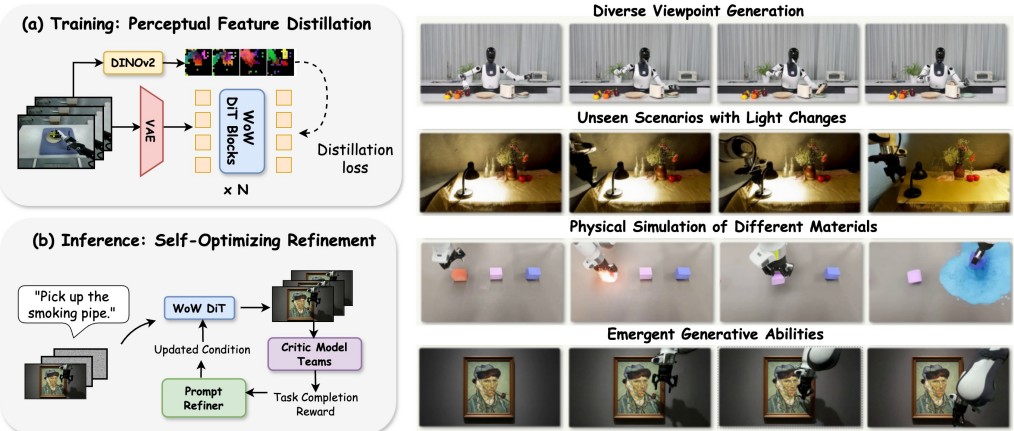

Figure 1: **WoW world model** is a world model system that integrates perception, imagination, self-optimization, and action. It learns from real-world data and generates high-quality, physically consistent robot action videos in out-of-distribution scenarios, enabling real-world robotic execution.

## Abstract

Generative models are pivotal for creating world models in robotics, yet they struggle to produce physically plausible dynamics, especially in complex, contact-rich manipulation tasks. Conventional approaches of embodied world models for manipulation simulation are often limited by explicit physical constraints or insufficient scale, leading to poor generalizability of robot embodiments, materials, action, or environments. We introduce WoW, a 14-B parameter embodied world model, to demonstrate that scaling, when guided by key architectural innovations, can unlock a new level of physical plausibility in complex manipulation simulation. Our approach is twofold: (1) As a foundation, we ensure visual-level realism with a novel token distillation loss that grounds the model in the robust feature space of a pre-trained vision model (DINOv2). (2) Furthermore, we propose a conceptual framework, a self-optimization World Model, implemented as a dynamic instruction refinement system that allows the model to improve its physical predictions during inference continuously, thereby enhancing both physical realism and temporal consistency. WoW demonstrates a strong grasp of physical causality and collision dynamics across a challenging set of 600+ manipulation videos with 4 core abilities and 20 sub-dimension tasks, on both human evaluation and metrics, and a 5-task real-world Franka evaluation. Our extensive scaling experiments reveal that performance on the most challenging, contact-rich tasks shows accelerated gains with larger training datasets. WoW sets a new state-of-the-art in generalizable manipulation simulation, producing physically plausible outcomes for tasks far exceeding the capabilities of previous generative models. We include our video demos and codes in wow-world-model-iclr.github.io

## 1 Introduction

How does a human child acquire an understanding of the world? Not by passively observing videos, but through active and physical experimentation. The cognitive scientist Jean Piaget succinctly articulated this principle: *"To know an object is to act on it" (Piaget, 2013)*. This form of embodied

learning, where countless 'actions' are intrinsically linked to immediate 'outcomes', is the foundational mechanism for mastering the laws of physics. This principle finds its direct computational instantiation in embodied world models (Zhen et al., 2025), which are explicitly designed to learn a predictive model of how the world responds to an agent's actions (Hafner et al., 2019).

In contrast, many recent advances in predictive models, particularly in video generation, are predicated on passive observation, a principle fundamentally distinct from active experimentation that fosters accurate causal understanding (Kang et al., 2025)). While models like Sora (Brooks et al., 2024) and others (Wan et al., 2025; Agarwal et al., 2025) achieve stunning photorealism and demonstrate emergent physical intuition, this intuition remains brittle. Their training objective prioritizes modeling statistical correlations from internet-scale data (Bain et al., 2021; Nan et al., 2025) over inferring the underlying causal mechanisms of physics. This superficial understanding of physical laws manifests as a series of catastrophic technical failures in robotic manipulation tasks that require precise physical reasoning: models fail to preserve object integrity during contact, exhibit systematic collision detection failures in multi-body interactions, and produce physically impossible dynamics with deformable materials and fluids. These are not mere rendering artifacts, but fundamental deficiencies in the generative model's ability to learn physical dynamics rather than just appearance. This leads to robot policies learned from fictitious physics being unreliable and unsafe in the real world.

To address these fundamental limitations, we introduce **WoW**, a 14-billion-parameter video DiT-based world model that ensures physical plausibility by learning from large-scale data (Peebles & Xie, 2022), rather than relying on explicit physical constraints or hand-engineered physics engines. Our approach incorporates two key architectural innovations. To inject the inherent physical properties of pre-trained models, we propose a novel token distillation loss that enhances visual realism by grounding the model in robust feature representations from DINOv2 citepOquab2023DINOv2. Furthermore, to ensure the reliability of the generated results, we introduce a *Self-Optimization world model* that refines language instructions during inference to maintain long-horizon prediction reliability and temporal consistency. This enables the achievement of physical fidelity during training and ensures accuracy verification during testing.

To validate WoW, we conduct a comprehensive experimental evaluation on 600+ robot videos from open-source datasets(Wu et al., 2024; Bu et al., 2025) and an in-house collection, which are designed to probe physical consistency and causal reasoning across 4 core abilities and 20 sub-tasks. On this benchmark, we answer five central Research Questions(RQ). We begin by establishing the foundational power of our model (**RQ1**), where our 14B-parameter WoW achieves state-of-the-art performance against all baselines, scoring an unprecedented 80.16% on physical law adherence and 96.53% on instruction understanding. Crucially, we demonstrate through extensive human evaluation that autonomy evaluation metrics are highly correlated with human preference.

We then probe the true depth of its learned physical knowledge, using WoWBench's diverse scenarios to test its ability to generalize to novel tasks (**RQ2**) and its capacity for abstract, counterfactual reasoning (**RQ4**). Finally, we demonstrate its transformative impact on intelligent agents. We show it can serve as a cognitive sandbox for a large vision-language model to debug its own plans (**RQ3**), and culminate our investigation by bridging the simulation-to-reality gap, translating its generated futures tp tangible manipulation on a physical robot (**RQ5**). Taken together, these experiments provide the first holistic and numerically-backed evidence that a generative video model can function as a powerful, generalizable, and practical world model for embodied intelligence.

## 2 RELATED WORK

Our work is forged at the nexus of two competing paradigms in world modeling, visually summarized in fig. 2. On one side stand models that learn compact, abstract representations (Fig. 2b), either for latent-space control as pioneered by the Dreamer series (Hafner et al., 2019; 2024), or for semantic understanding via non-generative prediction, exemplified by JEPAs (Assran et al., 2023; 2025). On the other side are large-scale generative models (Fig. 2a) like Sora (Brooks et al., 2024), which pursue high-fidelity pixel-level simulation but are plagued by a brittle grasp on physics, leading to critical failures in physical coherence and temporal consistency (Motamed et al., 2025; Zhen et al., 2025; Bansal et al., 2025). We include more related work discussion in section B.

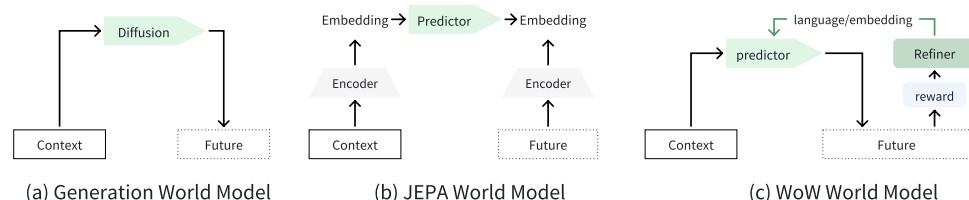

(a) Generation World Model      (b) JEPA World Model      (c) WoW World Model

Figure 2: **Comparison of Diffusion, JEPA (Assran et al., 2025), and WoW.** The **Predictor** generates a **Future** from the input **Context**. This outcome is then evaluated to produce a **reward**, which directs the **Refiner**. Finally, the **Refiner** leverages this reward and external **language/embedding** guidance to issue a corrective signal, iteratively improving the next prediction cycle.

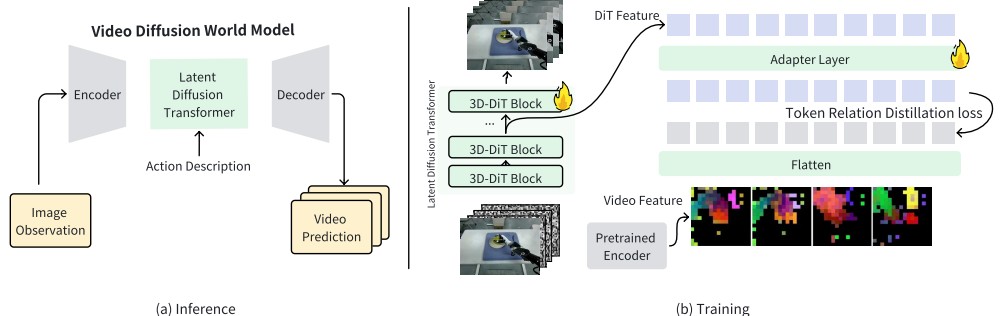

(a) Inference      (b) Training

Figure 3: **Overview of the Video Diffusion World Model.** (a) Inference: a latent diffusion transformer predicts future frames from image observations and text-based action descriptions. (b) Training: self-supervised learned encoder features supervise intermediate DiT representations via a token relation distillation loss to improve spatial-temporal modeling.

## 3 WoW World Model: Omni Self-Optimization Framework

We introduce WoW, an embodied world model built upon a novel test-time self-optimization conceptual framework. WoW is capable of iterative self-correction through a closed loop of imagination and reasoning.

Inspired by Neisser's Perceptual Cycle (Neisser, 1976), the framework of WoW structures this loop into three stages. WoW operationalizes this cycle through three corresponding, tightly integrated modules:

- **Task Imagination (Schemata):** The generative core of WoW. This is realized by a high-fidelity, physically-aware Diffusion Transformer that serves as the engine for pixel-level future prediction (see section 3.1).

- **Self-Optimization Reflection (Perception):** The self-correction mechanism of WoW. In WoW, this is implemented as a Solver-Critic agent system that verifies the physical consistency of imagined outcomes and iteratively refines them (see Section 3.2).

- **Behavior Extraction (Action):** The final stage that grounds imagination in reality. In WoW, this is a test-time module that translates the verified, imagined trajectories into executable robotic policies (see Section E.3).

### 3.1 Physically-Grounded Video World Model

The core of our system is a latent video diffusion transformer, trained to predict future video sequences from an initial state and a textual instruction. Its ability to generate physically plausible rollouts stems from a combination of architectural innovations and a novel supervision signal.

**Multimodal-Conditioned Diffusion Transformer.** The model's backbone is a DiT architecture (Peebles & Xie, 2022), conditioned on multimodal inputs. A pre-trained T5 encoder (Raffel et al., 2020) processes textual instructions (describing the task, environment, and embodiment), and

its embeddings are injected into the DiT via adaptive LayerNorm (adaLN) to guide the generation process. To imbue the model with strong spatio-temporal priors, we employ a hybrid positional encoding scheme: **absolute 3D positional embeddings** maintain global scene coherence, while **relative 3D RoPE** enforces local, pixel-level causality crucial for modeling contact dynamics.

**Spectral Decomposition for Motion Modeling.** To efficiently capture complex physical interactions, we process video inputs using a **3D Haar wavelet transform**. This decomposes the video into a low-frequency band (coarse scene structure) and high-frequency sub-bands (fine-grained motion details). This spectral separation allows the model to allocate its capacity more effectively toward modeling critical, high-frequency events like collisions and deformations, which are often lost in standard downsampling schemes.

**Structural Supervision via Relational Feature Distillation.** To transcend the limitations of pixel-level reconstruction losses, which often fail to enforce physical consistency, we introduce a novel structural supervision signal. We employ a pre-trained DINOv2 model (Oquab et al., 2023) as a "perceptual teacher." As shown in fig. 3(b), we extract intermediate representations from our DiT and apply a **Token Relation Distillation loss** against the features from DINOv2. By forcing our model to replicate the *relational structure* of the teacher's feature space, we explicitly teach it to understand object boundaries, spatial configurations, and contact dynamics. This endows the world model with a strong physical inductive bias.

**Structural Supervision via Relational Feature Distillation.** A core limitation of conventional VGMs is their reliance on pixel-level reconstruction losses (e.g., L1/L2). While effective for visual fidelity, these objectives often fail to enforce physical plausibility, leading to artifacts like object deformation or penetration. To overcome this, we introduce a novel training paradigm that moves beyond pixel supervision to directly supervise the model's internal representations.

Our key insight is to distill knowledge from a powerful, pre-trained self-supervised model, DI-NOv2 (Oquab et al., 2023), which has already learned rich semantic and structural priors about the physical world. Instead of merely using its features as input, we employ it as a "perceptual teacher" to guide the learning process of our DiT backbone.

As illustrated in fig. 3(b), during training, we extract the intermediate feature representations from a 3D-DiT block. These features are passed through a lightweight adapter layer to align their dimensionality with the features extracted from the corresponding video clip by the DINOv2 encoder. Crucially, we then apply a **Token Relation Distillation loss**. This loss penalizes the difference in the pairwise relational matrices between the DiT's tokens and the DINOv2's tokens. By forcing our model to replicate the *relational structure* of the teacher's feature space, we are not just matching values, but are explicitly teaching the DiT to understand object boundaries, spatial configurations, and contact dynamics. This deep, structural supervision endows our world model with a strong physical inductive bias, enabling it to generate videos that are not only visually coherent but also physically grounded.

## 3.2 SELF-OPTIMIZATION CLOSED-LOOP VIA SOLVER-CRITIC AGENTS

Building upon the conceptual foundation of WoW, we detail the *Experience Self-Optimization Reflection (Perception)* stage, a key component of our closed-loop mechanism. We introduce a solver–critic paradigm that enhances the physical consistency and realism of generated outputs. Unlike the limited abilities of prior work (Soni et al., 2024; Chi et al., 2025a), WoW employs a comprehensive agent system comprising a World Model, a Refiner Agent, and a Dynamic Critic Model Team, which together establish an iterative closed-loop workflow. An overview of this architecture is illustrated in Figure 4.

**Solver: WoW-DiT World Models.** The world model generates candidate videos from high-level user prompts and optional visual context. It functions as the solver, producing initial outputs that capture semantic intent but may not yet satisfy physical constraints.

**Refiner Agent.** The Refiner enhances prompt quality through iterative rewriting driven by critic feedback. By systematically incorporating physical constraints and task-specific details, it transforms underspecified instructions into precise, executable prompts.

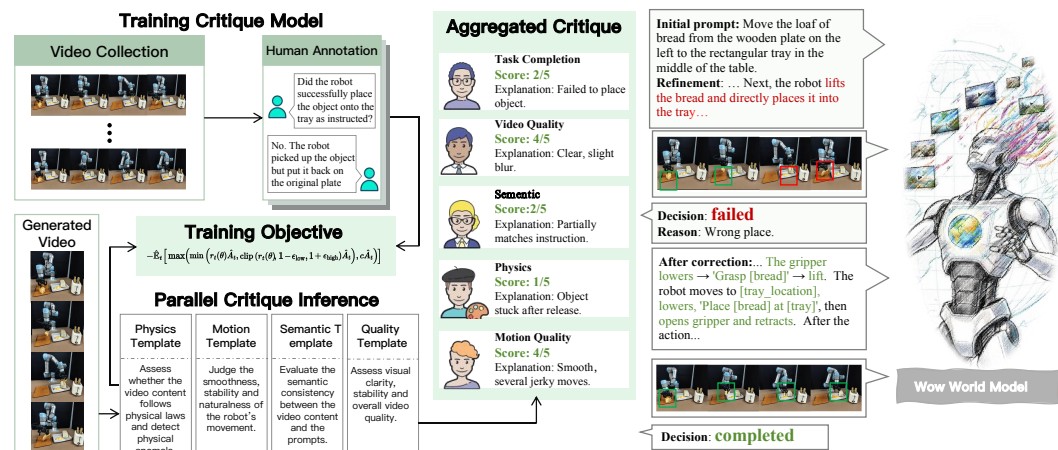

Figure 4: **Overview of Solver-Critic Video Generation Agents.** The framework consists of a world model that generates candidate videos from an initial prompt and video/image, a dynamics critic trained on annotated real and synthetic data to evaluate physical plausibility, and a prompt refiner that iteratively revises prompts based on critic feedback. This closed-loop system continuously improves video generation quality until physically consistent and satisfactory outputs are produced.

**Dynamic Critic Model Team.** Acting as the verifier, the critic assesses generated videos across dimensions such as task completion, physical plausibility, and kinematic smoothness. Fine-tuned from a vision–language model backbone, it delivers structured feedback rather than single scalar metrics, enabling precise and actionable prompt refinement.

**Closed-Loop Workflow.** These components form an adaptive solver–critic cycle: the world model generates, the critic evaluates, and the refiner revises prompts until both semantic coherence and physical consistency are achieved. This iterative process progressively converges on robust and realistic video outputs. A more detailed description of each component and an extended discussion connecting our design to the Prover–Verifier (Kirchner et al., 2024) paradigm are provided in Appendix E.2.

## 4 EXPERIMENTS

We conduct a comprehensive evaluation of proposed WoW model to address five central Research Questions: **(RQ1)** How does WoW compare against state-of-the-art video generation based world models, and what scaling laws govern its performance? **(RQ2)** To what extent does WoW generalize to novel tasks, embodiments, and physical scenarios unseen during training? **(RQ3)** Can WoW generate diverse futures, follow even counterfactual instructions? **(RQ4)** Can WoW work as a sandbox, generate long-horizon tasks as a simulator for VLM planners? **(RQ5)** Can the generative capabilities of WoW translate into practical, real-world robotic manipulation success?

### 4.1 EXPERIMENTAL SETUP

**Training Data.** Our model is trained on a meticulously curated dataset of unprecedented scale and diversity, comprising 2.03 million video clips (over 7,300 hours). This data, sourced from 5275 tasks over 200 procedurally generated scenes and featuring 12 distinct robot embodiments, is designed to instill a generalizable, embodiment-agnostic understanding of physical principles. A detailed description of the dataset and our rigorous filtering pipeline is provided in section E.1.

**Evaluation Benchmarks.** We evaluate all models on 606 video samples, the test benchmark designed to assess physical plausibility across tasks of varying difficulty (*Easy*, *Medium*, *Hard*). The classification, based on object properties, action complexity, task duration, and environmental factors, yielded 231 *Easy*, 237 *Medium*, and the remaining samples as *Hard*. Performance is measured using a suite of metrics, including autonomous evaluation scores, human preference scores, and task-specific success rates.

Table 1: **Comparative analysis of foundational video generation models.** We benchmark our **WoW-DiT** against SOTA models using direct text-to-video generation. All metrics: higher is better. Best results are **bold** with highlight.

| Model | Base | Human Evaluation | | | | | Autonomous Evaluation | | | | |
|---|---|---|---|---|---|---|---|---|---|---|---|
| | | VQ | IF | PL | Plan | Overall | VQ | IF | PL | Plan | Overall |
| Cogvideo | cogvideo | 3.29 | 1.52 | 1.73 | 1.30 | 7.84 | 18.93 | 5.91 | 44.13 | 2.32 | 10.34 |
| Cosmos-Predict1 | cosmos1 | 2.84 | 2.60 | 2.41 | 2.49 | 10.34 | 12.39 | 45.29 | 16.85 | 7.47 | 16.30 |
| Wan2.1 | wan | 3.49 | 1.79 | 2.30 | 1.62 | 9.21 | 12.92 | 12.63 | 5.96 | 5.6 | 8.59 |
| Cosmos-Predict2 | cosmos2 | 3.18 | 2.33 | 2.31 | 1.62 | 9.21 | 34.92 | 16.52 | 28.76 | 6.67 | 18.24 |
| *Our Foundational Model* | | | | | | | | | | | |
| **WoW-DiT** | cosmos1 | 3.12 | 2.86 | 2.78 | 2.84 | 11.60 | 61.76 | 74.82 | 40.38 | 2.89 | 27.10 |
| **WoW-DiT** | wan | **4.09** | 2.60 | **3.16** | 2.52 | 12.37 | **82.33** | 56.21 | 68.18 | 4.74 | 34.97 |
| **WoW-DiT** | cosmos2 | 3.76 | **3.19** | 3.03 | **3.36** | **13.34** | 65.60 | **78.53** | **80.25** | **6.88** | **41.07** |

Table 2: Autonomous evaluation of models with a self-optimization framework, using agents for refinement.

| Model | Base | VQ ↑ | IF ↑ | PL ↑ | Plan ↑ | Overall ↑ |
|---|---|---|---|---|---|---|
| cosmos1 + Agent | cosmos1 | 3.34 | 48.26 | 5.05 | 8.23 | 9.05 |
| cosmos2 + Agent | cosmos2 | 52.79 | 98.00 | 73.47 | **11.77** | 45.99 |
| **WoW + Agent** | **cosmos1** | **95.69** | 94.02 | **81.63** | 4.26 | 42.06 |
| **WoW + Agent** | **wan** | 92.56 | 6.73 | 75.91 | 6.75 | 23.77 |
| **WoW + Agent** | **cosmos2** | 75.26 | **96.53** | 80.16 | 7.76 | **46.11** |

Table 3: Data scaling law comparison in PBench.

| Data | PBench | | |
|---|---|---|---|
| | VLM ↑ | Qual. ↑ | Overall ↑ |
| 30k | 0.3901 | 0.3323 | 0.3612 |
| 200k | 0.5920 | 0.3790 | 0.4855 |
| 600k | 0.6240 | 0.3914 | 0.5077 |

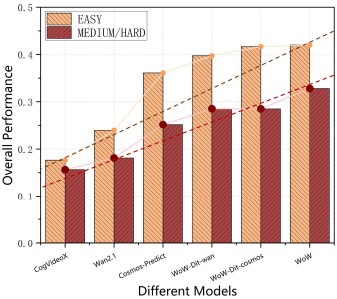

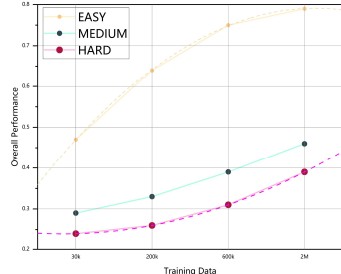

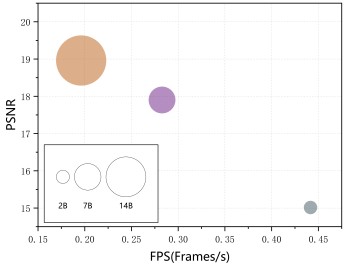

Figure 5: **Performance Comparison Across Different Models in WoWBench.** The solid lines with markers represent actual performance, while the dashed lines indicate the linear fitted trends for each difficulty level. * denotes models post-trained on our dataset.

Figure 6: **Scaling Curves for Training Data.** We divide the benchmark into three levels of difficulty: Easy, Medium, and Hard. The left figure shows that as training data increases from 30k to 2M, performance on the Easy tasks begins to saturate, while the Hard tasks continue to benefit from more data.

Figure 7: **Visual Quality Comparison Among scaling Model Size.** An analysis of inference speed and performance for models of varying sizes, specifically 2B, 7B, and 14B parameters. Performance is evaluated using the low-level metric, PSNR.

## 4.2 RQ1: PERFORMANCE SCALING AND SOTA COMPARISON

**Comparative Analysis.** We first benchmark WoW against leading video generation models. As shown in Table 1, our approach, **WoW-DiT**, consistently and substantially outperforms all baselines across both human and autonomous evaluation metrics, particularly in instruction following (IF) and physical law (PL) adherence. Notably, WoW-DiT built upon the Cosmos2 base model and achieves the highest overall scores, demonstrating the synergistic benefit of our proposed training methodology and a strong base architecture. The lower part of the table further reveals that our full **WoW** agent with self-optimization condition achieves the highest overall autonomous evaluation score (46.11), showcasing the power of the Solver-Critic loop.

**Scaling with Data and Model Size.** We empirically validate that WoW adheres to neural scaling laws. As illustrated in Figure 6, performance scales predictably with both data volume (from 30k to

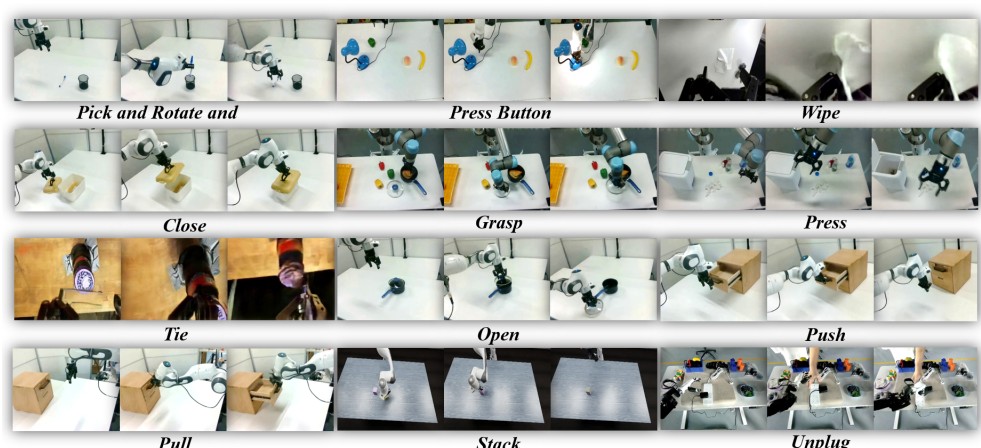

Figure 8: **More Generalization Ability** Case Study in generalization ability of five other aspects.

2M samples) and model size (from 2B to 14B parameters). Crucially, while performance on *Easy* tasks begins to saturate, gains on *Hard* tasks remain significant, suggesting that complex physical reasoning abilities are unlocked at larger scales and are far from their performance ceiling. This trend is corroborated in Table 3. Figure 7 highlights the inherent trade-off between performance and inference speed, a critical consideration for practical deployment.

**Impact of Architecture and Pre-training.** Figure 5 analyzes performance as a function of model architecture and pre-training data size. The results reveal a clear positive correlation: more recent architectures trained on larger datasets yield better performance. The differing slopes of the fitted trend lines for each difficulty level are particularly insightful. The steeper slope for *Easy* tasks ($y = 0.051x + 0.1567$) compared to *Medium* or *Hard* tasks ($y = 0.032x + ...$) quantitatively confirms that improving performance on complex, long-horizon physical reasoning is a significantly harder challenge that benefits less from generic model improvements alone, underscoring the necessity of specialized methods like our relational distillation.

### 4.3 RQ2: GENERALIZATION TO NOVEL SCENARIOS

A key desideratum of a foundational model is the ability to generalize beyond its training distribution. We assess this through extensive qualitative and quantitative evaluations.

**Qualitative Visualizations.** Figure 8 showcase WoW's remarkable zero-shot generalization in action instruction. More qualitative results is include in section C. The model successfully executes commands across unseen *embodiments* (from industrial UR5 arms to the dexterous Tiangong hand), a diverse *task repertoire* of 15 distinct skills, and profound *domain shifts* (e.g., executing tasks in sketch or oil painting styles). This provides compelling visual evidence that WoW learns an abstract and compositional understanding of physics, decoupling action from specific context.

**Quantitative Analysis of Physical Properties.** We quantitatively measure generalization across diverse physical phenomena in Table 4. WoW-Cosmos emerges as the clear top-performer, achieving state-of-the-art scores across nearly all categories, including rigid bodies, soft bodies, fluids, and elasticity. Its superior performance, especially on challenging non-rigid dynamics (e.g., highest PhyGen score on Soft body tasks), indicates a more accurate and robust internal representation of complex physical laws.

### 4.4 RQ3: WoW AS A COGNITIVE SANDBOX FOR VLM PLANNERS

We posit that a world model's ultimate value lies not in generation, but in serving as a **cognitive sandbox**—an internal world for an agent to simulate, plan, and self-correct. To prove this, we designed a complex task ("separate and stack colored cubes") where even a powerful VLM like Qwen-2.5-VL-7B (Bai et al., 2025) fails due to planning ambiguity.

Table 4: Comparative analysis of world models on physics simulation and visual benchmarks. Due to the number of metrics, the table is split into two parts for readability. The best score in each category is highlighted in **bold** and with .

| Model | Rigid | | | | | Soft | | | | | Fluid | | | | |
|---|---|---|---|---|---|---|---|---|---|---|---|---|---|---|---|
| | FVD | PhyGen | WMB | DreamSim | EQS | FVD | PhyGen | WMB | DreamSim | EQS | FVD | PhyGen | WMB | DreamSim | EQS |
| WoW-CogVideoX | 75.1 | 72.3 | 70.1 | 68.9 | 71.5 | 70.2 | 68.1 | 66.5 | 67.3 | 68.0 | 65.4 | 63.2 | 61.9 | 64.1 | 62.8 |
| WoW-SVD | 80.3 | 78.5 | 77.9 | 79.1 | 78.8 | 76.5 | 75.1 | 74.3 | 77.0 | 75.4 | 71.0 | 69.8 | 68.2 | 70.1 | 69.5 |
| WoW-Wan | 82.5 | 81.0 | 80.2 | 83.1 | 81.7 | 79.8 | 78.5 | 77.1 | 80.4 | 79.0 | 75.3 | 74.1 | 72.5 | 76.2 | 74.9 |
| WoW-Cosmos | **91.2** | **90.5** | **89.8** | **92.3** | **90.9** | **88.6** | **87.9** | **86.5** | **89.1** | **88.2** | **84.7** | **83.2** | **81.6** | **85.0** | **83.5** |

| Model | Gravity | | | | | Optics | | | | | Elasticity | | | | |
|---|---|---|---|---|---|---|---|---|---|---|---|---|---|---|---|
| | FVD | PhyGen | WMB | DreamSim | EQS | FVD | PhyGen | WMB | DreamSim | EQS | FVD | PhyGen | WMB | DreamSim | EQS |
| WoW-CogVideoX | 78.5 | 76.1 | 75.3 | 77.2 | 76.8 | 60.1 | 58.9 | 55.2 | 59.3 | 57.7 | 68.2 | 67.1 | 65.4 | 66.8 | 66.5 |
| WoW-SVD | 83.2 | 81.9 | 80.5 | 82.4 | 82.0 | **85.5** | **84.3** | **83.1** | **86.0** | **84.9** | 75.1 | 74.3 | 73.0 | 74.8 | 74.2 |
| WoW-Wan | 85.6 | 84.0 | 83.1 | 86.2 | 85.1 | 81.3 | 80.1 | 79.5 | 82.4 | 81.0 | 78.4 | 77.2 | 76.5 | 78.0 | 77.5 |
| WoW-Cosmos | **93.4** | **92.8** | **91.5** | **94.0** | **93.1** | 84.1 | 82.9 | 81.7 | 85.2 | 83.5 | **87.3** | **86.5** | **85.8** | **88.1** | **87.0** |

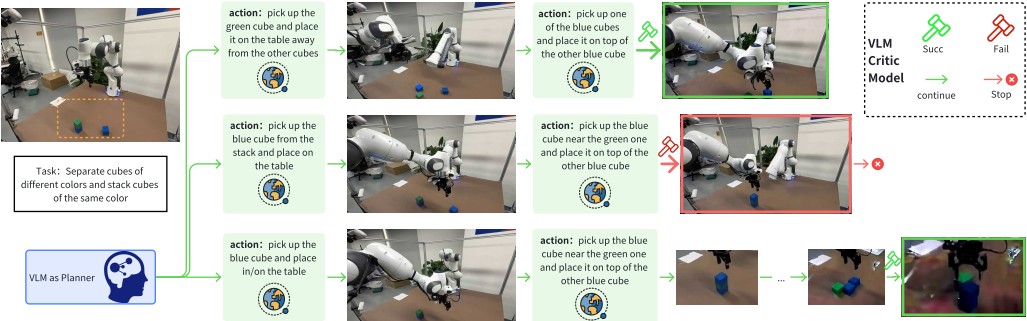

Figure 9: **Self-correction of VLM planning via world model simulation.** (a) Our iterative loop: a VLM planner proposes an action, our world model simulates the future frame, and a VLM critic provides feedback, enabling the planner to refine its next step. (b) Terminal frames from the simulation, illustrating a successful plan (top) versus a detected failure (bottom) that triggers re-planning.

We then implement a test-time feedback loop where the VLM "thinks" by interacting with our world model (Figure 9). The results are stark: without interaction, the VLM's planning accuracy is a mere 33%, leading to a **0% task success rate**. After just two rounds of simulated reflection within WoW, its planning accuracy skyrockets to **89%**, which directly translates into a **44% task success rate**. This leap from zero to success, achieved purely through internal simulation, is definitive proof that WoW provides the crucial grounding for an AI to debug its own logic. It transforms a brittle planner into a robust problem-solver, establishing generative world models as a foundational component for truly intelligent agents.

## 5 RQ4: CAN WoW GENERALIZE TO COUNTERFACTUAL PHYSICAL CASES?

To test for physical reasoning beyond simple pattern replication, we challenge the model with text-based counterfactuals that alter a scene's physical laws. The model must interpret the new abstract rule and generate a physically coherent simulation of its consequences.

When presented with the counterfactual that a block is "*impossibly heavy*", the model does not generate a simple failure state. Instead, it simulates the **physics of failure**: the gripper strains and the arm shows tension, while the block remains inert on the table (Figure 10). This demonstrates reasoning from a new, abstract principle, not the recall of a trained example. This capability is confirmed across a range of counterfactuals (see demo page), validating the model's capacity for out-of-distribution physical reasoning.

### 5.1 RQ5: EXAMINING WoW IN REAL-WORLD ROBOTIC MANIPULATION

To bridge the sim-to-real gap, we developed an Flow-Mask Inverse Dynamics Model (IDM) as describe in section E.3 that translates WoW's generated videos into executable robot actions (see Appendix [X]). This model establishes our system's physical ceiling, achieving a **94% action replay**

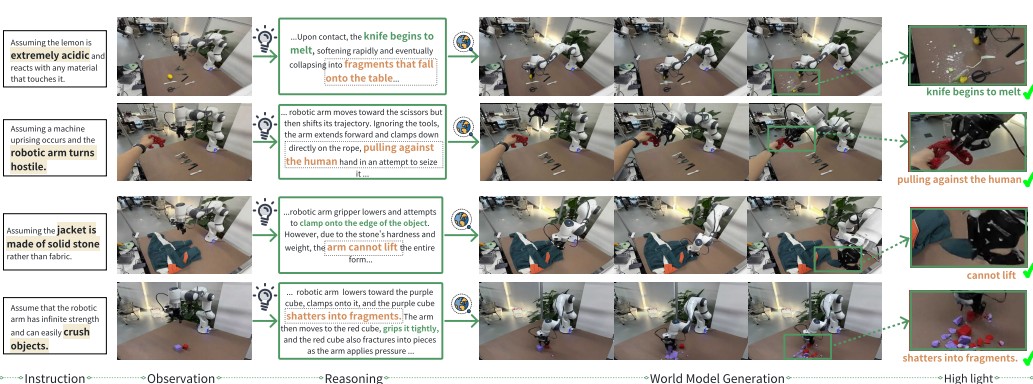

Figure 10: **Counterfactual Physical Reasoning.** The model receives a scene and a textual counterfactual altering a physical law ( "jacket is made of stone"). It correctly interprets the new rule and generates a simulation of the physical consequences—visualizing the inability to lift an impossibly heavy object. This grounds an abstract linguistic rule in a dynamic physical prediction.

**accuracy**, which represents the upper bound for task success. We then deployed WoW's plans onto a physical robot for a series of manipulation tasks (Figure 11).

The results are stark: models without fine-tuning ('w/o FT') struggle, validating the difficulty of real-world deployment. In contrast, fine-tuning provides a quantum leap in performance. Our premier model, **WoW-cosmos2 with FT, achieves a success score of 0.64**, decisively outperforming all baselines. This proves WoW captures a sufficiently accurate model of physics to guide a physical robot, transforming abstract goals into successful real-world actions.

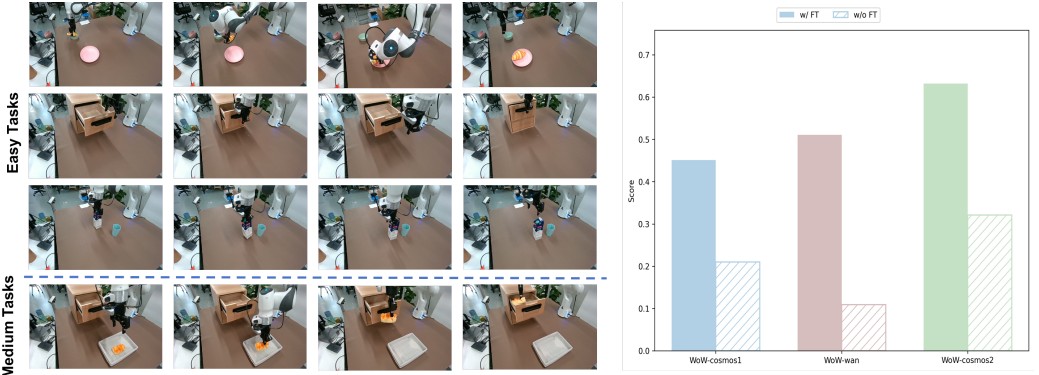

Figure 11: **WoW's Efficacy in Real-World Robotics. (Left)** Qualitative examples of successful trajectories generated by WoW for *easy* and *medium* difficulty tasks executed on a physical robot. **(Right)** Quantitative results demonstrating the real-world accuracy comparison of three different world model backbones. Across all base models, fine-tuning provides a dramatic boost in real-world performance, with WoW-cosmos2 achieving the highest score.

# 6 CONCLUSION

This work confronts the critical limitations of passively-trained video models in robotics. We introduced WoW, a 14B-parameter world model that learns robust physical dynamics from large-scale interaction data, not just statistical correlations. Through architectural innovations like token distillation and a self-optimization framework, WoW achieves a superior grasp of causality. Our comprehensive evaluation demonstrates SOTA performance in physical reasoning and generalization.

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

## A    THE USE OF LLMS

Large language models (LLMs) were consulted for writing guidance; LLMs polished the manuscript, and we further employed LLMs to refine the prose and enhance the overall exposition.

## B    RELATED WORK

Our work is forged at the confluence of three powerful currents in world model research: latent dynamics modeling for control, abstract predictive representation learning, and large-scale generative simulation.

**Latent Dynamics and Predictive Representations.**    The foundational lineage of world models, pioneered by Ha and Schmidhuber (Ha & Schmidhuber, 2018), established the principle of learning a compact predictive model in a compressed latent space. This was significantly advanced by recurrent state-space models (RSSMs) for planning (Hafner et al., 2018) and culminated in the Dreamer series (Hafner et al., 2019; 2024), which demonstrated scalable control by learning policies entirely within a learned latent "dream." A parallel paradigm, Joint-Embedding Predictive Architectures (JEPAs) (Assran et al., 2023; Bardes et al., 2024), learns powerful semantic representations by predicting masked information in an abstract embedding space, yielding strong priors for robotics (Assran et al., 2025) without pixel-level generation. Our work affirms the value of abstract representations but argues for the necessity of pixel-level simulation for grounding complex physical interactions.

**Generative Video World Models.**    The recent advent of large-scale video models like Sora (Brooks et al., 2024) has shifted the frontier towards creating high-fidelity "neural simulators." These models, whether autoregressive (Hu et al., 2023; Bruce et al., 2024) or diffusion-based (Yang et al., 2023), excel at visual synthesis. However, they suffer from critical limitations for embodied intelligence, including physical incoherence (Motamed et al., 2025; Chow et al., 2025; Bansal et al., 2025), a lack of 3D and temporal consistency (Zhen et al., 2025), and often require specialized memory architectures to maintain long-horizon coherence (Hong et al., 2024; Chai et al., 2023; Zhou et al., 2025).

**Our Contribution.**    WoW synthesizes these threads. We harness the simulation power of diffusion-based video models while directly addressing their physical and structural deficiencies. Instead of relying on explicit physics engines, we instill a deeper understanding of physical relationships through knowledge distillation from a pretrained foundation model, aiming to create a scalable, learnable, and generalizable generative physical engine for embodied intelligence.

## C   CASE STUDY: EMERGENCY ABILITY

### C.1   MORE GENERATION VISUALIZATION

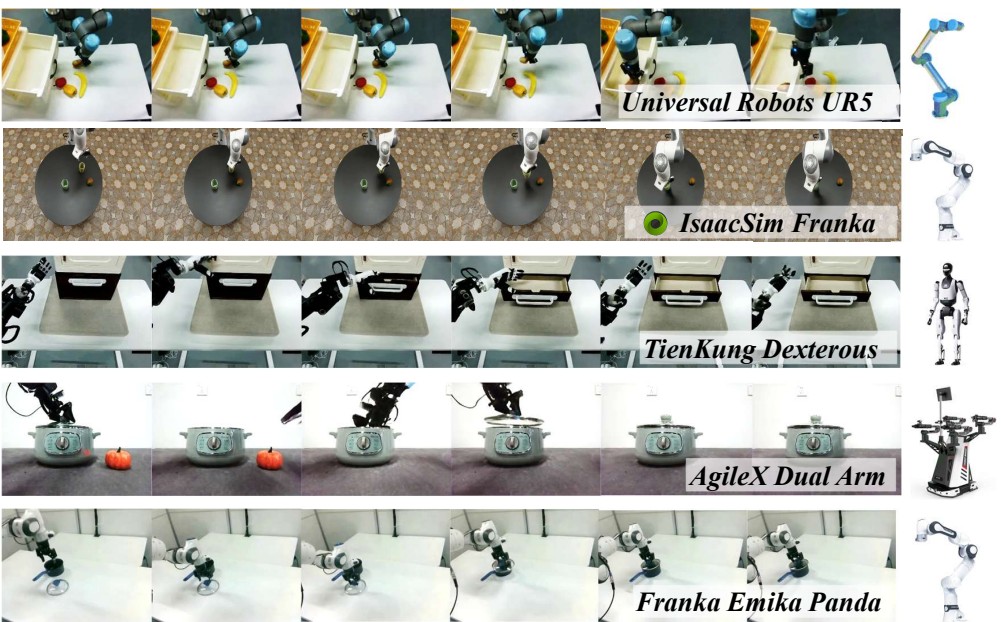

Figure 12: **Cross-Embodied Generalization Ability** Case Study in generalization ability of different robot types.

### C.2   COUNTERFACTUAL REASONING AND REPLANNING

This module evaluates the model's capacity to generalize its planning and video generation under counterfactual assumptions. Starting from a baseline scenario (see Figure 10), the robot is tasked with grasping a blue block, lifting it, placing it stably, and returning to its initial position. The generated video shows the successful execution of this plan in a controlled laboratory setting.

We then introduce explicit counterfactual modifications to the textual prompt. For instance, by assuming that "*the blue block is extremely heavy and far beyond the robot's lifting capacity,*" the regenerated description and video depict the robot gripper tightly closing around the block, its joints straining under tension, yet the block remains immovable on the tabletop. This shift demonstrates the model's ability to adaptively reconcile linguistic assumptions with physical constraints, producing trajectories consistent with the altered premise rather than repeating the baseline motion.

Altogether, we design nine counterfactual conditions, ranging from altered material properties (e.g., the blue block as a water-soaked sponge, or the tabletop and gripper being unusually slippery), to modified environmental dynamics (e.g., gravity shifting to a 45-degree angle, or the arm moving clumsily and misaligned), and extreme physical phenomena (e.g., block replication, strong inter-block attraction, or time freezing near the target). These variations constitute both **depth tests**, where the baseline scene is perturbed with controlled counterfactuals (see Figure 13), and **breadth tests**, where the same mechanism is applied to diverse scenes with randomized counterfactual prompts (see Figure 10).

The results indicate that the model not only accommodates specific counterfactual constraints, but also generalizes them across contexts, revealing robustness in trajectory adaptation and a promising capacity for systematic out-of-distribution reasoning.

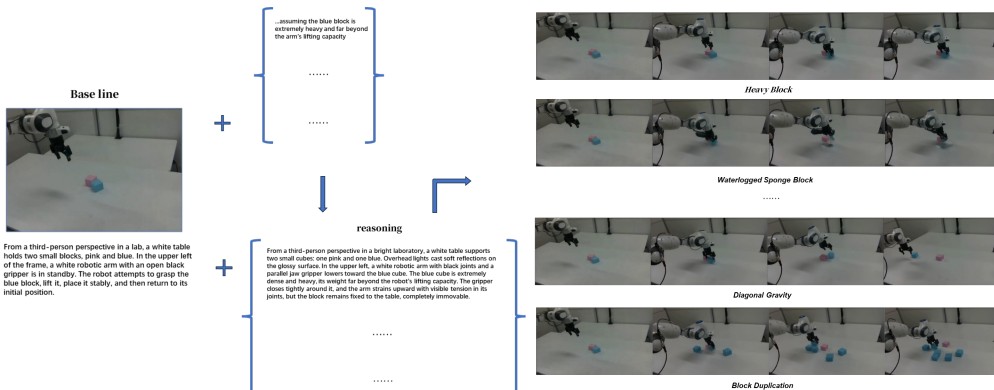

Figure 13: Counterfactual reasoning with depth tests.

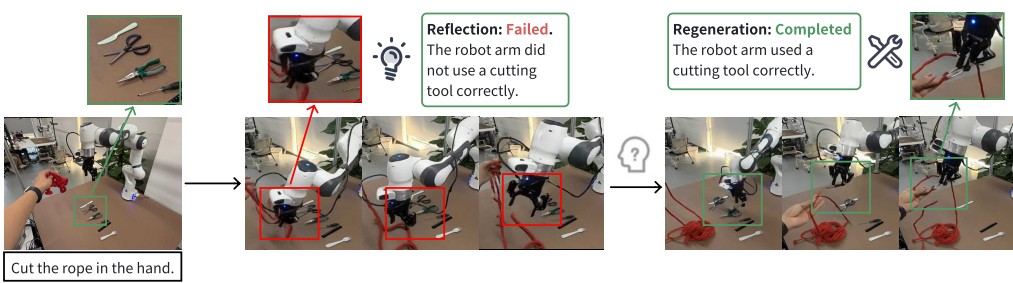

Figure 14: Case study illustrating emergent creativity and self-reflection in a rope-cutting task.

## C.3 EMERGENT CREATIVITY AND SELF-REFLECTION

We conduct a case study on a rope-cutting task to test the model's capacity for both creative problem-solving and self-reflection. The overall process, as illustrated in Figure 14 , begins with a short prompt "*Cut the rope in the hand.*" and an initial frame. In the first attempt, the generated video shows the robot directly cutting the rope using its manipulator without employing the appropriate cutting tool. Subsequently, the VLM judge evaluates the video and identifies that "*Failed. The robot arm did not use a cutting tool correctly.*". This feedback then guides the self-refine for regeneration. After regeneration, the new video shows the robot successfully using scissors to cut the rope, thus completing the task with the correct tool. This case demonstrates that our model has reflection capability, enabling it to creatively explore alternative solutions, correct execution errors, and improve task reliability. More importantly, it also reveals the model's emergent generalization to OOD tasks.

## C.4 PHYSICAL AND LOGICAL CONSTITUTIONALITY

This section presents a concise case study (see Figure 15)demonstrating how our two-stage *Logical Parsing → Action Instruction Rewriting* mechanism grounds language containing negation and conditionals into executable action sequences. First, for the logical negation instruction "clear the tabletop, leaving only the blue objects behind." the VLM, using the initial frame, detects that the tabletop contains two screwdrivers and one blue tool, and normalizes the negated description as: Remove = {two screwdrivers}, Keep = {blue tool}. It then produces a linear plan (grasp each screwdriver in turn, place it into the drawer, then reposition the retained blue tool), avoiding common end-to-end failures such as leaving one removable item or mistakenly removing the blue tool. Second, for the conditional instruction "If the drawer is open, take out the blue cube; otherwise, knock the drawer three times." the VLM first determines the drawer's open/closed state from the initial frame: if open, it rewrites the prompt into a two-step sequence (grasp the blue cube; lift it out of the drawer); if closed, it rewrites it into an approach plus triple-knock sequence, eliminating the mixed behaviors (simultaneously attempting to open/knock/grasp) often observed when the condi-

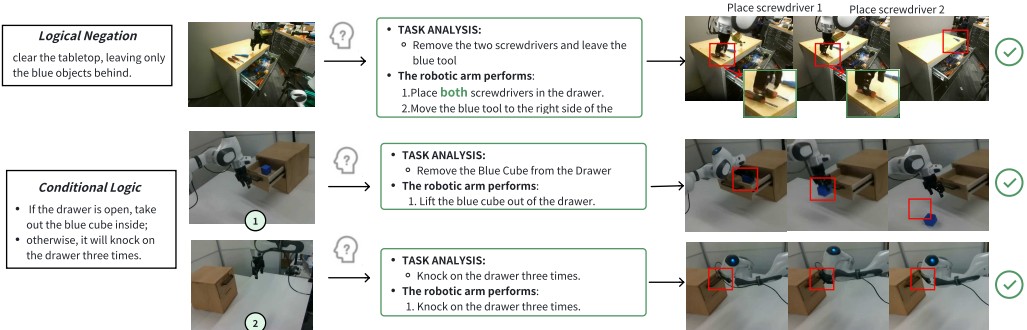

Figure 15: **Compositional Reasoning.** The figure illustrates two examples, showing WoW's reasoning ability. Specifically **Logical Negation** (top) and **Conditional Logic** (bottom), which grounds symbolic reasoning in imagined physical interactions.

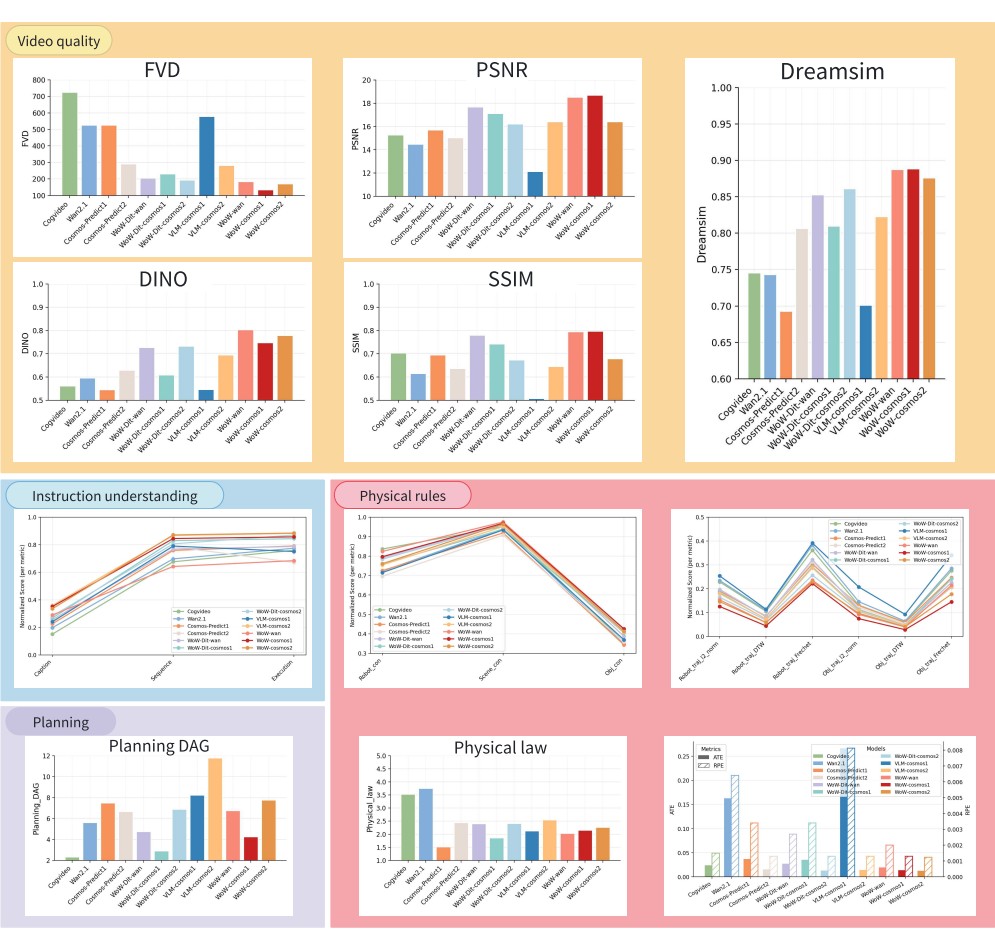

Figure 16: **Performance Comparison Across Different Models in Detail Metrics in benchmark.** Different color blocks stand for different dimensions in WoWBench. In every block, we demonstrate intuitive charts to present detailed scores in varied metrics in our WoWBench.

tion is unresolved by an end-to-end model. The three illustrated sub-scenarios (negation plus the two conditional branches) show that explicit Task Analysis and atomized action listing provide the video generation/control module with a clear target set and ordering constraints, yielding executions that strictly adhere to the linguistic logic, whereas the original complex prompts rarely achieves.

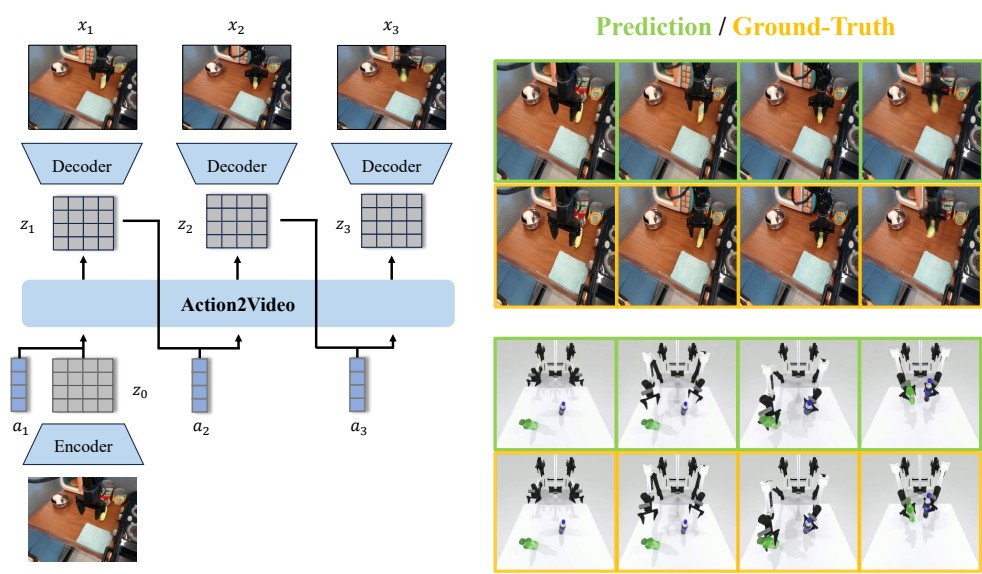

Figure 17: **Inference procedure of Action2Video** Action2Video trains a latent video diffusion model in the latent space provided by pre-trained variational autoencoder (VAE) in SDXL. An autoregressive transformer is used to predict future tokens conditioned on the corresponding action at each step.

# D  DOWN STREAM TASK APPLICATION

## D.1  ACTION-TO-VIDEO GENERATION FOR ROBOT MANIPULATION

A critical limitation of existing world models for robot manipulation lies in their inability to accurately capture **fine-grained robot–object interactions** from textual descriptions. This stems from the inherent **modality gap** between natural language and video frames (). In contrast, state-of-the-art robotic policies such as **DP**(Chi et al., 2023), **ACT**(Zhao et al., 2023), and **OpenVLA**(Kim et al., 2024) represent behavior through dense action trajectories that specify end-effector positions, orientations, and gripper states. While text-to-video models may appear as a potential inspiration, they primarily rely on high-level textual cues rather than frame-specific action instructions, and thus fail to model robot control with the required temporal precision.

We introduce **Action2Video**, a framework for generating **high-resolution (up to 640×480)** and **long-horizon (over 240 frames)** videos of robot manipulation directly from action trajectories. Following DiT (Peebles & Xie, 2023), Action2Video employs a **diffusion-based spatial–temporal transformer backbone** to capture complex environmental dynamics. To explicitly align actions with their visual consequences, we integrate a **frame-level action-conditioning module** into each transformer block. This design enables precise correspondence between actions and generated frames, supporting both **successful and failed rollouts** and modeling fine-grained control such as end-effector rotations. Furthermore, **autoregressive rollout** allows Action2Video to generate long-horizon, temporally consistent videos.

**Definition.** We define the *trajectory-to-video task* as predicting a video of a robot executing a trajectory $a_{t:t+n}$, given a sequence of historical observation frames $O_{t-h:t}$. Formally, $a_t \in \mathbb{R}^d$ denotes the action at timestep $t$, where $d = 7$ for a typical robot arm: three DoFs for translation, three for rotation, and one for gripper control. Figure 17 illustrates the overall inference pipeline of Action2Video and presents the prediction results on both single-arm and dual-arm robot datasets. The conditioning input is

$$c = \{z_{t-h:t}, a_{t:t+n}\}, \quad z_{t-h:t} = \text{Enc}(x_{t-h:t}), \tag{1}$$

and the diffusion target is the latent representation of the subsequent video frames

$$x_{t+1:t+n+1} = \text{Dec}(z_{t+1:t+n+1}). \tag{2}$$

# E MODEL IMPLEMENTATION

## E.1 PRETRAIN DATA PREPARATION

We construct our training dataset through a multi-stage pipeline designed to ensure both quality and diversity. The process consists of four sequential stages: **Collection**, **Filtering**, **Refinement**, and **Rebalancing**. Unlike simply enlarging the dataset with indiscriminate samples, our approach emphasizes that *data quality plays a decisive role in model performance*, and carefully curated data prove more effective than raw scale (Collaboration et al., 2023; Khazatsky et al., 2024; Radford et al., 2021; Northcutt et al., 2021; Zhang et al., 2017; Luo et al., 2024).

**Collection.** We collect thousands of hours of videos from multiple robotic platforms, including Agibot (Bu et al., 2025), Droid (Khazatsky et al., 2024), Robomind (Wu et al., 2025), and a large amount of in-house data. These sources cover a variety of embodiments and task scenarios, providing broad coverage across environments and robot types. This diversity serves as the foundation for building generalizable robot learning datasets.

**Filtering.** The collected data are processed through a series of filtering rules. Only RGB videos are retained, with BGR channels semi-automatically converted into RGB format for consistency. Static or non-informative sequences are removed, and a minimum length of 90 frames is enforced to ensure sufficient temporal context. In addition, we restrict the dataset to specific viewpoints such as head, wrist, and third-person perspectives, which best capture robot actions and task dynamics.

**Caption Refinement.** To further enhance the training signal, sparse textual annotations are expanded into dense descriptions using a pretrained VLM. Both uniform and sequential frame sampling are applied, ensuring coverage of both global context and local temporal transitions. Sparse and dense text annotations are combined with an approximate ratio of 1:4, and robot model identifiers are manually added into the text metadata. This step improves both the richness of supervision and the alignment between visual and textual modalities (Radford et al., 2021; Lin et al., 2024).

These prompts include:

- **Environment description** (*Environment Description*)
- **Robot setup** (*Robot Setting*)
- **Action description** (*Action Description*)
- **Final state description** (*End State Destcription*)

This ensures the world model remains consistent and informative even when converting into video frames.

---

**Prompt Example**

**"The robot's primary goal is to place a blue LEGO brick into a drawer and close it.** The scene takes place in an office environment with dim artificial lighting. The robot is a single-arm manipulator with a black and metallic appearance, equipped with a parallel jaw gripper. The camera is positioned in a first-person perspective, likely mounted on the robot's arm or head, providing a direct view of the workspace.

The robot's sub-goal is to insert the blue LEGO brick into the drawer. The action sequence begins with the robot perceiving the blue LEGO brick on a WoWden surface. The robot executes a 'Grasp [blue LEGO brick]' action, lifting it slightly. It then performs a 'Move_arm_to [drawer_opening]' command, positioning the brick above the drawer. The robot proceeds with a 'Place [blue LEGO brick] at [inside_drawer]' action, carefully positioning the brick within the drawer. Following this, the robot executes a 'Close_drawer' action, ensuring the drawer is securely shut.

After the action, the blue LEGO brick is neatly placed inside the closed drawer, and the robot's gripper is open and retracted from the drawer area.

---

**Rebalancing.** Finally, we address imbalance across tasks by increasing the sampling probability of underrepresented tasks. This ensures that rare but important skills are not neglected during training, and improves robustness across diverse robotic behaviors (Northcutt et al., 2021).

**Training Data Summary** Our training dataset was meticulously curated to provide a rich and diverse foundation for learning a generalizable world model. It comprises 2.03 million video clips, totaling over 5,000 hours of interaction footage, which corresponds to approximately 1 billion frames sampled at a consistent 24 frames per second. To foster robust generalization, the data were collected from over 200 procedurally generated simulated scenes, spanning contexts from complex household environments (e.g., kitchens, living rooms with cluttered objects) to structured industrial settings (e.g., warehouses, assembly lines). Crucially, the dataset features a diverse collection of 12 distinct robot embodiments to ensure the model learns a wide range of physical dynamics and morphologies.

The collection is dominated by industrial manipulators, with a significant emphasis on both single-arm and dual-arm configurations. The primary data sources include trajectories from the dual-arm Franka FR3, the single-arm UR5e, and the dual-arm UR5e, which together constitute a substantial portion of the dataset. To further broaden the diversity, we also incorporate data from various other platforms, including multiple Franka Emika Panda setups and several specialized configurations from the ARK, AgileX, and Tienkung series, ensuring a broad spectrum of kinematic properties and action spaces are represented. For pre-processing, all video sequences were captured at a native resolution of 640×480 and subsequently upsampled to 720×1024 pixels to align with our model's architectural input requirements. We applied a rigorous filtering pipeline to ensure data quality, which led to the exclusion of approximately 75% of the initial raw data. This high discard rate was a deliberate choice to remove trajectories with simulation instabilities, severe collisions, task failures, and periods of static inactivity, thereby ensuring the final dataset consists of high-quality, meaningful interactions.

Through this pipeline, we construct a dataset that is large in scale, carefully curated, temporally consistent, and densely annotated with semantic and physical labels—providing a robust foundation for training advanced robot learning models.

### E.2 CLOSED-LOOP IMAGINATION VIA SOLVER-CRITIC AGENTS

Building upon the conceptual framework of WoW, this section details the Experience Reflection (Perception) stage—a key component of our closed-loop mechanism. We introduce a solver-critic paradigm that enhances the physical consistency and realism of generated outputs. Unlike the Simple ability of previous work (Soni et al., 2024; Chi et al., 2025a), WoW has a comprehensive agent system, driven by a Refiner (Section E.2.2), which guided by a Dynamic Critic Model Team (Section E.2.3), iteratively improves the generated content. This dynamic workflow forms a Closed-Loop Generative Workflow (Section E.2.4), ensuring that the model's output is continuously refined and verified.

#### E.2.1 FRAMEWORK OVERVIEW

Achieving physically plausible video generation for complex, long-horizon robotic tasks requires moving beyond unidirectional models to a closed-loop, agentic system capable of self-perception and optimization. We frame this generative process as a deliberative act, analogous to the interplay between the intuitive "System 1" and the analytical "System 2" cognitive modes (Weston & Sukhbaatar, 2023). In our framework, an initial video serves as a "proposal" (System 1), which is then subjected to a rigorous critique and refinement loop that embodies the structured reasoning of System 2.

This architecture transforms the model from a passive generator into an active problem-solver. Our solver-critic framework is built upon three core components: the Refiner Agent, which optimizes the input and generate video output; the Dynamic Critic Model, which evaluates the generated output; and the integrated closed-loop Workflow. Furthermore, we discuss how this architecture aligns with the Prover-Verifier paradigm (Kirchner et al., 2024), showcasing its potential to endow the generative process with a new level of cognitive depth.

### E.2.2 REFINER AGENT IN WORLD MODEL

The quality of a generative model's output is highly dependent on its input prompt. For video generation in specialized fields like robotics, prompts must capture subtle physical details to produce plausible outcomes. However, manually crafting such high-quality prompts is a time-consuming and arduous process of trial and error. While the emerging field of Automatic Prompt Engineering offers systematic optimization methods for language tasks (Khan et al., 2025; Agrawal et al., 2025), these general approaches are not directly tailored to the unique demands of physically-grounded video synthesis.

To address this challenge, we introduce the Refiner Agent, an autonomous system designed for test-time prompt optimization that does not require retraining the underlying video generation model. The agent takes a high-level user instruction and initiates an iterative refinement loop. In each iteration, a dedicated prompt rewriting module enhances the prompt's specificity and physical consistency. This rewriting process is explicitly guided by structured feedback from our Critic Model Team (Section E.2.3), which identifies errors or missing details, such as adding constraints to prevent objects from passing through solid surfaces. Conceptually, this iterative process performs a guided search over the discrete prompt space, where the critic feedback functions as a "textual gradient" (Pryzant et al., 2023; Yuksekgonul et al., 2024). Our approach thereby transforms prompt engineering from a manual, trial-and-error task into a systematic, data-driven, closed-loop optimization process.

### E.2.3 DYNAMIC CRITIC MODEL TEAM

Functioning as the 'verifier' in our iterative refinement loop, the Dynamic Critic Model Team is the second core component of our system. The need for this specialized critic arises because traditional metrics such as Fréchet Video Distance (FVD), Peak Signal-to-Noise Ratio (PSNR), and Structural Similarity Index Measure (SSIM) (Unterthiner et al., 2018; Huynh-Thu & Ghanbari, 2012; Hore & Ziou, 2010), while capable of assessing visual fidelity, are inadequate for evaluating physical realism—a critical bottleneck in the development of robust world models. To address this gap, we align with the emerging consensus that VLMs are the cornerstone of next-generation video assessment (Chen et al., 2024). However, general-purpose VLMs lack the domain-specific precision required for tasks like robot manipulation. We therefore construct our specialized critic by fine-tuning a VLM on a curated Question-Answering (QA) dataset containing both real and model-generated videos of robotic operations. This dataset is structured to probe the model's understanding across five key dimensions: task completion, action success, physical plausibility of interactions (e.g., stability, deformation), kinematic smoothness, and overall quality. This targeted fine-tuning transforms the generalist VLM into a reliable expert verifier, instilling it with the specialized knowledge required to accurately assess the physical dynamics of robot interaction.

### E.2.4 CLOSED-LOOP GENERATIVE WORKFLOW

As illustrated in Figure 4, our system integrates the Refiner Agent and Dynamic Critic Model into a closed-loop workflow that transforms video generation from a single-pass operation into an iterative refinement process. The loop initiates with a high-level user task, which the Refiner Agent expands into a detailed, physically-constrained prompt for our generation model (WoW). The resulting candidate video is then evaluated by the Dynamic Critic Model for physical plausibility and semantic coherence. If the video is judged 'incomplete' or 'failed', the critic provides structured feedback that the Refiner Agent incorporates to revise the prompt for the subsequent generation cycle. This iterative process of generation, critique, and refinement reframes video synthesis as an adaptive reasoning task. By endowing the generative pipeline with this self-corrective capability, our workflow enables the system to progressively converge on outputs that constitute a robust, physically grounded world model.

### E.2.5 DISCUSSION: THE PROVER-VERIFIER PARADIGM FOR GENERATIVE WORLD MODELS

To understand our architecture on a deeper level, this section explicitly connects it to established theoretical frameworks in the field of artificial intelligence—namely, the Solver-Critic (Wang et al., 2025a; McAleese et al., 2024; Gou et al., 2023) and Prover-Verifier (Kirchner et al., 2024) paradigms. In these paradigms, one agent (the Prover/Solver) is responsible for generating a candi-

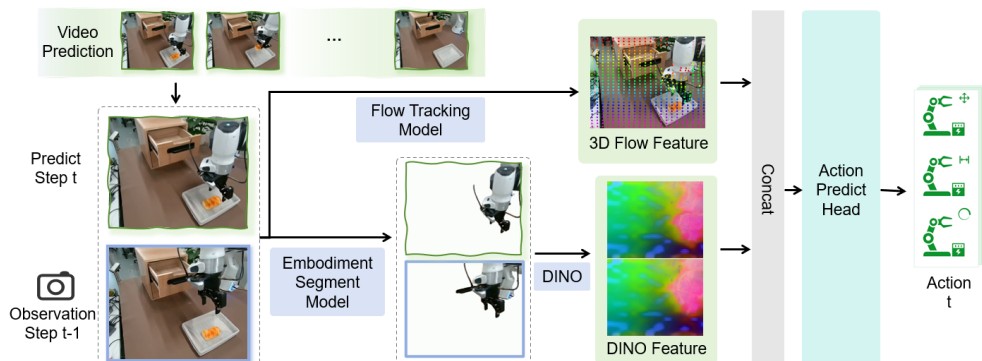

Figure 18: Work flow of inverse dynamic model. Giving two frame predictions, our FM-IDM can estimate the delta End-Effect action of the robot.

date solution, while another, often simpler or more specialized agent (the Verifier/Critic), is responsible for evaluating its correctness.

Our architecture provides a concrete implementation of the established Prover-Verifier and Solver-Critic paradigms (Wang et al., 2025a; McAleese et al., 2024; Gou et al., 2023; Kirchner et al., 2024). Within this framework, the Refiner Agent function as the Prover/Solver, responsible for proposing and iteratively refining candidate videos. The Dynamic Critic Model Team acts as the Verifier/Critic, tasked with evaluating the physical plausibility of these proposals. A key contribution of our work is being the first to successfully apply this paradigm-traditionally used for discrete, logical tasks such as mathematical proofs (Lin et al., 2025) and code generation (Wang et al., 2025b)-to the high-dimensional, continuous, and stochastic domain of video generation.

The primary advantage of this approach is its ability to optimize for complex, non-differentiable objectives like "physical realism" without requiring an explicit, differentiable loss function. The Prover learns to generate outputs that are accepted by the Verifier, providing a powerful mechanism for instilling abstract values like physical common sense into generative models. To summarize, this framework paves the way for building physically and causally consistent world models suitable for robotics planning.

### E.3 FLOW-MASK INVERSE DYNAMICS MODEL

The proposed FM-IDM is a video-to-action model that maps predicted video frames to real-world robot execution transitions. Instead of relying on model-specific features (Liao et al., 2025; Chi et al., 2025b; Hu et al., 2025), we adopt a pixel-level decoding approach, trading real-time performance for greater generality and accuracy (Ko et al., 2023; Tan et al., 2025). Designed as a plug-and-play module, our model is compatible with any visual generative world model, enabling system-level evaluation and facilitating reward extraction via embodied interaction.

**Task Formulation**    Given two consecutive visual observations $(o_t, o_{t+1})$ from the predicted video — each corresponding to the underlying robot states $(s_t, s_{t+1})$ — the goal is to infer the end-effector action $a_t$ that transitions the robot from $s_t$ to $s_{t+1}$. The inverse dynamics model $F_\delta$ takes the current frame $o_t$ and the corresponding flow $\mathcal{F}_{t \to t+1}$ as input, and outputs a predicted delta action $\hat{a}_t$:

$$\hat{a}_t = F_\delta(o_t, \mathcal{F}_{t \to t+1}) \tag{3}$$

**FM-IDM**    To achieve this, we first estimate a motion field $\mathcal{F}_{t \to t+1}$ capturing the geometric transformation between frames. The estimated flow encodes both translational and rotational motion of the manipulator. We implement $F_\delta$ as a two-branch encoder-decoder network. We first fine-tuned a SAM (Kirillov et al., 2023) that process the masked current frame $o_t$ to extract scene and embodiment context; the other processes the optimal flow by CoTracker3 model (Karaev et al., 2024) $\mathcal{F}_{t \to t+1}$ to capture fine-grained temporal dynamics, as described in Figure 18. In conjunction with the with the DINO (Oquab et al., 2023) feature, we further use Multi-Layer Perceptron (MLP) as

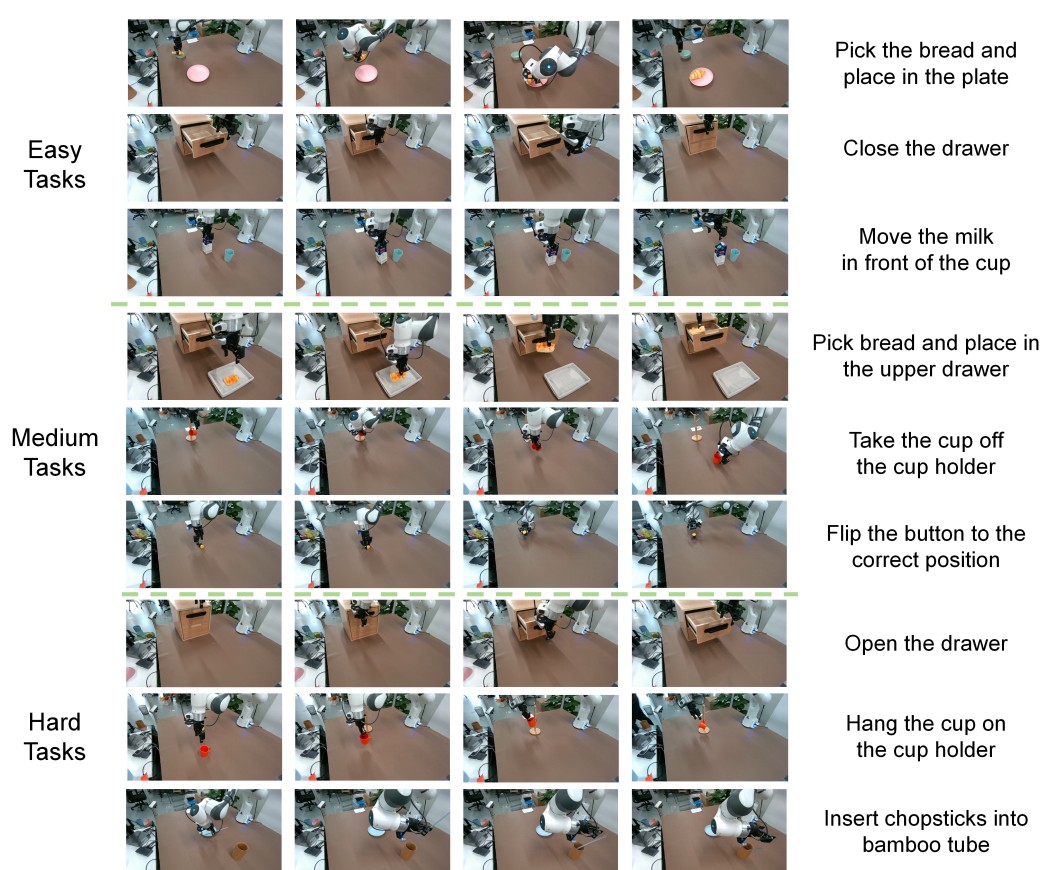

Figure 19: **Difficult Level Separate of IDM.**

action head to learn the 7-DoF action feature. The training objection is as follows:

$$\min_{\delta} \ \mathbb{E}_{(o_t, o_{t+1}, a_t)} \ d\big(a_t, F_\delta(o_t, \mathcal{F}_{t \to t+1})\big) \tag{4}$$

where $d(\cdot, \cdot)$ denotes a weighted smooth L1 loss in the end-effector action space.

By explicitly modeling spatio-temporal correspondences, the model generalizes better across diverse tasks, background variations, and occlusions, and is robust to noise in video-based prediction.

**Embodiment-Centric End-Effector Action Dataset**  To facilitate the learning of end-effector actions directly from visual input, we curate a dataset of 646k image–action pairs across 219 tasks, covering a broad range of manipulation scenarios. The dataset is carefully constructed to span a diverse action space and densely cover the reachable workspace of the robot, ensuring that the model learns from a comprehensive set of physically plausible end-effector configurations. More details of the implementation are included in Section 4.

**Real-World Feedback through IDM**  At the action execution stage, rewards are grounded in physical feasibility and obtained through direct interaction with the environment. They may be defined in multiple ways: binary success/failure of task completion, distance-based metrics between predicted and actual end-effector positions, force/torque stability measures during contact, or energy-efficient motion profiles. Failures serve not merely as penalties but as corrective feedback that guides continual adaptation. Importantly, this reward can be further fed back to the world model, adjusting the model through Group Relative Policy Optimization (GRPO) for evolutionary visual generation (Xue et al., 2025).

