# OpenReview forum: "WoW: Scaling Embodied Omni-World Model For Generalizable Manipulation Simulation"
_ICLR.cc/2026/Conference — Submitted to ICLR 2026_

### Official Review · Reviewer_tziz · 2025-10-27

**Soundness:** 3
**Presentation:** 3
**Contribution:** 4
**Rating:** 6
**Confidence:** 2

**Summary:**

This paper introduces WoW, a 14-billion-parameter embodied world model designed to overcome the challenge of generating physically plausible dynamics in complex,contact-rich manipulation tasks. The authors propose two core innovations:

1) A novel perceptual feature distillation loss which makes the models' visual-level realism align with DINOv2;
2) A self-optimization conceptual framework implemented as a Solver-Critic agent system during inference, which dynamically and iteratively refines language instructions to continuously enhance physical prediction accuracy and temporal consistency.

According to the paper, a comprehensive evaluation on 600+ manipulation videos, including human and autonomous metrics, and a 5-task real-world evaluation, demonstrates that WoW has great comprehension about physical causality and collision dynamics.

**Strengths:**

1. Novel and crucial architectural innovations: The work presents a highly innovative approach by integrating two architectural pillars: Perceptual Feature Distillation and a Self-Optimization Solver-Critic Framework. The distillation explicitly injects physical priors into the model while the Solver-Critic loop iteratively refines prompts with structured feedback, significantly enhancing the output's physical consistency and semantic coherence at inference time.
2. Robust generalization and abstract reasoning capabilities: WoW demonstrates strong generalization across unseen robot, diverse tasks, and profound domain shifts. On the other hand, the work shows an impressive capacity for abstract counterfactual physical reasoning by generating physically coherent videos based on linguistic rules that alter physical laws.
3. Extensive and wide-ranging experimental validation: The paper includes a comprehensive evaluation covering performance scaling laws(RQ1), generalization to novel scenarios (RQ2), utility as a cognitive sandbox for planners(RQ3), counterfactual reasoning (RQ4) and translation to real-world robotic execution (RQ5).

**Weaknesses:**

1. Although the Token Relation Distillation Loss is a core innovation for the work, the paper lacks a detailed ablation study on its independent contribution to performance, particularly on the PL metric. For instance, a direct comparison between WoW-DiT(with distillation) and WoW-DiT(without distillation). Besides, the specific mathematical form and implementation details of the Token Relation Distillation Loss and overall loss function for the Video Diffusion World Model are not explicitly provided, which hinders a thorough understanding of how structural information is encoded.
2. Figure 7 highlights the inherent trade-off between generative model performance and inference speed. Considering the WoW's 14B parameter count and the added complexity of the closed-loop critic component, it is necessary to provide a comprehensive comparison of inference speed(FPS) and execution latency against baseline models to realistically assess its instantaneous viability in a real robotic system.
3. The real-world performance evaluation is sparse. While Figure 11(Right) presents quantitative scores for different WoW world model backbones, it lacks horizontal comparison with other SOAT world models or classic embodied intelligence models(e.g., policy-learning methods). Additionally, the metric used in the histogram ("Score") is general and needs further clarification.

**Questions:**

1. Quantitative Contribution of Token Relation Distillation Loss:
    1. The Token Relation Distillation loss is a crucial innovation. Please provide a direct ablation study: What is the PL score in Table 1 and Table 4 for a WoW model that has not been trained with this distillation loss (i.e., using only standard reconstruction losses)?
    2. In Table 2, the performance score between cosmos2+Agent and WoW+Agent is not very significant. Does this marginal difference suggest a boundary effect where the perceptual feature distillation offers less significant gains, or potentially introduces minor interference, when applied to an already highly effective base model like cosmos2?
2. DINOv2 is used as the perceptual teacher. Have the authors experimented with other pre-trained models as a teacher? To what extent do different models have or are expected to have an impact on WoW?
3. The Self-Optimization closed-loop mechanism introduces computational overhead and latency due to multiple inference steps. How many iterations on average are typically required to reach the "Decision: completed" state? Are these accumulated delays acceptable for real-time planning in robotic systems?
4. Please explicitly define the "Score" shown in Figure 11 (Right). Does it represent the success rate? If so, is it the average success rate across the five evaluation tasks mentioned in the abstract? Furthermore, to fully assess WoW's potential, is it possible to include a horizontal comparison with at least one representative classical or state-of-the-art embodied intelligence model (e.g., BC or Diffusion Policy baseline) in the real-world setup?
5. Clarifications on Figures and Tables:
    1. In Table 2, please clarify if the "Agent" in the baseline models (cosmos1/2 + Agent) refers to the Self-Optimization framework proposed in this paper, or a different agent(VLM). Also, please correct the bold marking, as the **IF 98.00** for cosmos2 + Agent appears to be the true optimum.
    2. In Figure 5, the caption mentions:"\* denotes models post-trained on our dataset", but the asterisk symbol is missing from the figure and legend. Please clarify the intended usage or remove the mention.
    3. In Figure 9, the caption is confusing. The terminal frames should illustrate "a successful plan **(top)** versus a detected failure **(middle)** that triggers re-planning, leading to a corrected plan **(bottom)**," based on the visual flow. Please correct the description for accuracy.

---

### Official Review · Reviewer_uA8S · 2025-10-31

**Soundness:** 2
**Presentation:** 3
**Contribution:** 4
**Rating:** 4
**Confidence:** 4

**Summary:**

The paper introduces WoW, a prompt-able video diffusion world model for robot manipulation. It aims to generate physically plausible video simulations of complex, contact-rich robotic manipulation tasks without relying on an explicit physics engine. The paper identifies that standard video generation models often fail on physical accuracy.
WoW proposes two core mechanisms to achieve this:
1. **Token Relation Distillation** loss, which aims to align the model's internal representations with pre-trained DINOv2 features on top of standard pixel-level losses.
2. **Dynamic Instruction Refinement System** (with solver-critic-refiner), which critiques generated rollouts and iteratively rewrites instructions to enhance temporal consistency and task success. At inference time, the model uses a "Solver-Critic" framework. A generative model (the "Solver") produces a video, which is then evaluated by "Critics". Based on this feedback, a "Refiner" agent rewrites the text prompt to guide the model toward a more physically accurate generation in a subsequent iteration.

The authors evaluate the model on a new benchmark, WoWBench, and demonstrate improved performance in physical plausibility against other generative baselines. The paper reports higher instruction following and physical law adherence than prior text-to-video baselines and show steeper scaling gains on contact-rich tasks as data increase. They also conduct physical robot evaluation with Franka robots to verify effectiveness of WoW in real-world manipulation tasks.

**Strengths:**

- **Clarity**: The paper is well-written and clearly motivates the problem. The core problem (i.e., the lack of physical plausibility in large-scale video models and their subsequent usefulness for robot manipulation) is well-established, and the paper's proposed high-level approach is easy to understand.

- **Scale and Scope**: The paper addresses a large-scale problem: moving beyond passive video generation toward interactive, physically-grounded simulation. The work involves a significant engineering effort, resulting in a 14B parameter model and a new dataset, WoWBench. The authors provide an extensive set of experiments, including scaling laws, generalization tests, and quantitative physics benchmarks (Table 4). The paper also includes a real-world robot evaluation (RQ5) as a step toward validating the model's usefulness.

**Weaknesses:**

- **Limited Novelty in Core Contributions**: The core generative model is a standard Diffusion Transformer (DiT), and the "Token Relation Distillation loss" is a well-established knowledge distillation technique among works that aim to ground the model’s embedding with pretrained representation models. The paper fails to differentiate this from prior work.

- **Complicated and Artificial Critic Framework**: The "Self-Optimizing Refinement" loop appears heavily engineered. The framework uses multiple VLM critics fine-tuned on specific QA datasets for dynamically refining the action descriptions. However, considering the engineering effort, it seems like a more straightforward alternative would be to directly fine-tune the primary video diffusion model itself with specialized QA datasets, which may result in similar performance gain.

- **Insufficient Comparison to Key Baselines**: The paper claims itself as a video generation / world model framework that is effective for robotic manipulation, but authors do not compare against policy learning methods with latent world models (e.g., Dreamer). There is no evidence this computationally-intensive approach (which requires video generation, refiner-critic cycle, and FM-IDM)  is superior to simpler and more compact latent-space models for policy learning.

- **Questionable Conditioning and Subjective Emergent Abilities**: The evidence provided is insufficient to confirm that the model is precisely conditioned on text prompts, especially for subtle variations (e.g., slight directional changes), and that the model is capable of emergent counterfactual generation (Appendix C.2). The claim that modifying a text prompt (e.g., declaring an object "extremely heavy") leads to a novel, physically-consistent outcome (e.g., the robot straining) without explicit counterfactual training is questionable. While the demo results can be visually convincing, the evaluation of such complex, emergent behavior remains highly subjective.

- **No Failure Modes**: It would be nice to provide analyses on failures modes of the proposed method to justify some design choices or mention potential rooms for improvements. Especially, since the physical robot experiments require multiple stages (video generation, refiner-critic cycle, and FM-IDM), it would be nice to analyze what component failed under which condition/situation.

- Nit: RQ4 (Section 5) should be probably subsection 4.4, and subsequently RQ5 (Section 5.1) should be subsection 4.5.

- Nit: Grammar in line 254 sounds awkward. Perhaps it should be "(RQ3) Can WoW generate diverse futures and follow even counterfactual instructions?"

- Nit: Some figures are stacked on top of each other, reducing the readability of the submission. Also, some tables are never explained in the main body of text (e.g., *The lower part of the table*, instead of *Table 2*, in line 319). Please consider rearranging/refactoring them.

**Questions:**

- **Justification for Critic Framework**: Why was the complex, iterative Solver-Critic framework (with multiple finetuned VLMs) chosen over the much simpler baseline of using the critic's specialized QA fine-tuning datasets to directly fine-tune the DiT world model? I would expect an ablation study comparing these two approaches, as the current design seems unnecessarily convoluted.

- **Evidence of Action-Conditioning**: Can the authors provide more rigorous qualitative or quantitative evidence of action-conditioning? For example, please show a grid of video generations where only a single key word in the action prompt is changed (e.g., "push red block to green target" vs. "push blue block to green target" vs. "push red block to blue target"). Is it also possible to provide more quantitative evidence behind counterfactual generation?

- **Comparison to Latent World Models**: One of the submission's weaknesses is the lack of comparison to SOTA latent world models (like Dreamer). Why was this line of work for policy learning (model-based RL/IL with latent dynamics) ignored? Without this comparison, the claim that WoW is a useful tool for manipulation is unsubstantiated.

---

### Official Review · Reviewer_rFph · 2025-10-31

**Soundness:** 2
**Presentation:** 2
**Contribution:** 2
**Rating:** 2
**Confidence:** 4

**Summary:**

The paper introduces a video-world model, named WoW, that improves physical feasibility of video generation by (1) improving internal representations (by aligning with DINOv2 features) in base video-generation model (2) iterates on prompt with a critic feedbacks (realism, adherence, motion quality, etc.) to refine generated video.

**Strengths:**

* The authors present an improved video generation model that is capable of producing more physically plausible motions for contact rich + deformable object manipulation.
* Authors evaluate the model on interesting axes of video generation like counterfactual scenario, soft, fluid, etc.,

**Weaknesses:**

* The writing could use some improvement. Many statements are imprecise, unsubstantiated, and use unwanted flowery language characteristic of LLM generation. While I’m not personally against LLM use, I believe it is imperative that the authors ensure that claims made are well supported. Here are some problematic segments:
  * Abstract (L033): “guided by key architectural innovations, …”
    * No architectural innovations were introduced in the paper.
  * Introduction (L052-L57): this segment talks about the importance of real world interaction in learning a dynamics model – which clearly the proposed video generation model does not have so I think of this segment as entirely irrelevant. L072 seems to suggest you address this limitation somehow which makes it an inaccurate statement to make.
    * The conclusion (Section 6, L481) can also make it explicit what limitations you intended to address precisely.
  * Section 2 – the related work could use much more improvement especially in terms of situating contributions (1) and (2) made in the paper (as listed in the summary above) better.
    * The current related work section just points to other world modeling attempts and says nothing about the key contributions introduced which have been experimented in other domains.
    * Clearly, the ideas claimed as novel in this paper (Section 3.1 L172-179, L185-198) have been explored for image generation (see weakness 2 below) and this should be acknowledged. The second contribution is also not novel, however its application in this case of video generation might be considered novel.
* The idea of improving internal representation was explored in recent literature like REPA\[1\], however there is no mention of it. In fact, the writing suggests that this is a novel contribution – if that’s the case the authors have to justify this better.
* The paper is missing a lot of key details in the main text, see questions below.

**Questions:**

- Was the 14B video generation model trained from scratch on the curated dataset? If not, what was the base model from which this was fine-tuned?
- What was the VLM backbone used to train the dynamics critic family of models? Where does the annotated human data for fine tuning these models come from?
- What model is the refiner agent is – it a closed-source API VLM model? How flexible is your video generation model in handling different language prompts that the refiner comes up with? Did you augment prompts when training the video generation model?
- Were any additional measures taken to ensure that the comparison to prior models in Table 1 is fair?
  - If your model is tuned better towards manipulation examples doesn’t it get an unfair advantage over models that have not?
  - How disparate are the test scenarios used for evaluation in Table 1 from those used in training – could you point to more qualitative evidence for this?
  - A fairer comparison to evaluate contribution (1) would be to fine-tune a base model like Cosmos with the dataset you used to train (/finetune?) your model.
- Table 4 does not contain any detail beyond ablations with different models? Could you add comparisons to existing baseline models on those benchmarks and just list your best performing backbone?
- Section 5.1 (RQ5): L431,L452 – not sure I understand what the authors mean by “system’s physical ceiling” and “94% action replay accuracy” – could the authors clarify what these terms actually mean and how to interpret them?

**References**
\[1\] Yu, Sihyun, et al. "Representation alignment for generation: Training diffusion transformers is easier than you think." *arXiv preprint arXiv:2410.06940* (2024).

---

### Official Review · Reviewer_ZeQc · 2025-11-01

**Soundness:** 2
**Presentation:** 2
**Contribution:** 3
**Rating:** 4
**Confidence:** 3

**Summary:**

This paper introduces WoW, a video generation world model that can be conditioned on text-based actions.
Its main contributions are:
- Applying feature distillation from a pre-trained vision model (DINOv2).
- Training a critique model and using it to improve the video generation quality.

The experiments focus primarily on evaluating video generation quality.

**Strengths:**

- The videos provided on the website appear visually photo-realistic.

**Weaknesses:**

- Except for Table 1, many experiments (such as Table 4 and Figure 11) compare variants of the proposed model with different base models, making it difficult to evaluate the effects of the proposed mechanisms.
- Please consider providing video demonstrations for the manipulation policy rollouts and direct comparisons between the proposed model and other models.
- Although the abstract lists DINO feature distillation as a main contribution, there is no clear ablation study providing an apple-to-apple comparison.

**Questions:**

- The overall presentation quality of the paper requires improvement. Both the writing and figures could be refined.
- The supplementary materials are not well organized, containing issues such as typos, empty folders, and video playback interruptions.
- Please introduce each metric (through a reference or textual description) rather than listing only abbreviations in the table.
- Wrap the website link in the abstract using a LaTeX `\url{}` command to make it more visible.

---

### Meta-Review · Area_Chair_9g2g · 2026-01-11

**Summary:**

This work proposes WoW, a large (14B) text/action-conditioned video world model aimed at generating physically plausible, contact-rich manipulation rollouts without an explicit physics engine. The reviewers have shown concerns on unclear novelty, incomplete ablations, and insufficiently rigorous/fair baseline comparisons, and there is no author responses. Thus the AC recommends rejecting the paper.

**Reviewer Concerns:**

see Summary

**Reviewer Scores:**

see Summary

---

### Decision · Program_Chairs · 2026-01-26

Reject